# Analysis of epidemiological association patterns of serum thyrotropin by combining random forests and Bayesian networks

**Ann-Kristin Becker**[1,2], **Till Ittermann**[3], **Markus Dörr**[2,4], **Stephan B. Felix**[2,4], **Matthias Nauck**[2,5], **Alexander Teumer**[3], **Uwe Völker**[2,6], **Henry Völzke**[2,3], **Lars Kaderali**[1,2]*, **Neetika Nath**[1,2]*

**1** Institute of Bioinformatics, University Medicine Greifswald, Greifswald, Germany, **2** DZHK (German Centre for Cardiovascular Research), Partner Site Greifswald, Greifswald, Germany, **3** Institute for Community Medicine, SHIP/Clinical-Epidemiological Research, University Medicine Greifswald, Greifswald, Germany, **4** Department of Internal Medicine B, University Medicine Greifswald, Greifswald, Germany, **5** Institute of Clinical Chemistry and Laboratory Medicine, University Medicine Greifswald, Greifswald, Germany, **6** Interfaculty Institute of Genetics and Functional Genomics, Department of Functional Genomics, University Medicine Greifswald, Greifswald, Germany

* lars.kaderali@uni-greifswald.de (LK); nikkihathi@gmail.com (NN)

**Data Availability Statement:** Data from the Study of Health in Pomerania (SHIP) cannot be shared publicly as they contain potentially identifying and sensitive medical information on study

## Abstract

### Background

Approaching epidemiological data with flexible machine learning algorithms is of great value for understanding disease-specific association patterns. However, it can be difficult to correctly extract and understand those patterns due to the lack of model interpretability.

### Method

We here propose a machine learning workflow that combines random forests with Bayesian network surrogate models to allow for a deeper level of interpretation of complex association patterns. We first evaluate the proposed workflow on synthetic data. We then apply it to data from the large population-based Study of Health in Pomerania (SHIP). Based on this combination, we discover and interpret broad patterns of individual serum TSH concentrations, an important marker of thyroid functionality.

### Results

Evaluations using simulated data show that feature associations can be correctly recovered by combining random forests and Bayesian networks. The presented model achieves predictive accuracy that is similar to state-of-the-art models (root mean square error of 0.66, mean absolute error of 0.55, coefficient of determination of $R^2 = 0.15$). We identify 62 relevant features from the final random forest model, ranging from general health variables over dietary and genetic factors to physiological, hematological and hemostasis parameters. The Bayesian network model is used to put these features into context and make the black-box random forest model more understandable.

participants. However, data access can be requested from the Forschungsverbund Community Medicine data access committee (online application form at http://fvcm.med.uni-greifswald.de) for researchers who meet the criteria for access to confidential data.

**Funding:** SHIP is part of the Community Medicine Research network of the University of Greifswald, Germany, which is funded by the Federal Ministry of Education and Research (grants no. 01ZZ9603, 01ZZ0103, and 01ZZ0403), the Ministry of Cultural Affairs as well as the Social Ministry of the Federal State of Mecklenburg-West Pomerania, and the network' Greifswald Approach to Individualized Medicine (GANI_MED)' funded by the Federal Ministry of Education and Research (grant 03IS2061A). Genome-wide data have been supported by the Federal Ministry of Education and Research (grant no. 03ZIK012) and a joint grant from Siemens Healthineers, Erlangen, Germany and the Federal State of Mecklenburg- West Pomerania. The University of Greifswald is a member of the Caché Campus program of the InterSystems GmbH. This work was further supported by the project Superthyreose, funded by the German "Innovationsfonds des Gemeinsamen Bundesausschusses" (grant no. VSF2_2019-167) AKB received funding from the BMBF (LiSyM, grant number 031L0032) and gratefully acknowledges an add-on-fellowship from the Joachim Herz Stiftung. LK acknowledges funding from the European Union (EuCanShare, grant 825903), as well as the State of Lower Saxony and the Volkswagenstiftung (Indira, grant number ZN3437). The funders had no influence on study design, data analysis, study interpretation, decision to publish, and writing of the manuscript.

**Competing interests:** The authors have declared that no competing interests exist.

## Conclusion

We demonstrate that the combination of random forest and Bayesian network analysis is helpful to reveal and interpret broad association patterns of individual TSH concentrations. The discovered patterns are in line with state-of-the-art literature. They may be useful for future thyroid research and improved dosing of therapeutics.

## Introduction

Machine learning (ML) based on epidemiological data offers an attractive approach to discover predictive patterns in complex biomedical systems, such as thyroid homeostasis. However, many ML models mainly aim at high predictive accuracy and do not offer easy model interpretability and explainability, which is often necessary for healthcare applications and research. Random forests (RF) [1] are such an example. They achieve high predictive accuracy by ensemble learning of a multitude of decision trees. However, in contrast to single decision trees, RFs may lead to a variety of complex decision paths, which makes them challenging to interpret. Therefore, they are usually considered black-box models. This work aims at proposing a workflow that puts focus on the interpretation of feature associations by complementing a black-box RF model with a post-hoc Bayesian network analysis.

Bayesian networks are probabilistic models describing (in-)dependence structures among random variables. They are well interpretable and offer an intuitive visualization of feature interactions. The presented workflow allows identifying predictive patterns from epidemiological and clinical data and permits it to visualize and understand the nature of these patterns. We apply the proposed workflow to data from the Study of Health in Pomerania (SHIP) [2] in order to identify broad predictive patterns of individual serum thyrotropin concentrations.

Thyrotropin, also known as thyroid-stimulating hormone (TSH) is a central component of the thyroid homeostatic system and a major diagnostic as well as an important therapy monitoring target in thyroid dysfunction. The main treatment goal of thyroid dysfunction is the renormalization of the thyroid function, mostly monitored by serum TSH concentration. However, its strong dependence on external factors makes it challenging to specify a generally valid optimum. The goal of our study was to discover and decipher association patterns of individual serum TSH concentrations from broad cohort study data to complement the study situation with a wholistic view. With a prevalence estimated at around 1–10%, thyroid dysfunction is one of the major endocrine disorders in Europe [3] and worldwide [4]. It causes a wide range of symptoms, including changes in the gastrointestinal system, heart rate, mood, skin, sexual function, and sleep. However, due to the mild or unspecific nature of these symptoms, thyroid dysfunction often stays undetected. Yet, it has been shown that even mild long-term imbalances of thyroid hormone levels increase cardiovascular risk, risk of dementia, and bone disorders, amongst others [5, 6]. TSH is produced by the anterior pituitary gland and stimulates the thyroid gland to secrete thyroxine (T4), which is then further converted to triiodothyronine (T3). Elevated levels of free T3 and free T4 in the blood plasma, in turn, inhibit the production of TSH via a negative feedback loop. Thus, serum TSH is a sensitive and easily accessible indicator of thyroid (dys-) function.

However, TSH is not steadily released from the pituitary gland but follows circadian and ultradian rhythms [7]. Moreover, TSH levels in serum fluctuate depending on life phases, reaching exceptionally high levels during periods of growth, stress, or pregnancy. Consequently, the individual TSH level depends on various external factors, including sex, age, diet,

or stress level [8]. The treatment goal in thyroid dysfunction is the renormalization of the thyroid function, monitored by the TSH level, but the optimum seems to be highly individual and may even be genetically predetermined [9–11]. TSH serum concentrations are thus hard to interpret, depend on a patient's general status und reference ranges are constantly under debate [8, 12, 13]. With TSH being a central marker and treatment target for thyroid dysfunction, there is considerable interest in investigating patterns associated with the individual TSH concentration in serum. Identified patterns may be highly valuable for therapeutic decision-making.

The relation of TSH to other thyroid hormones is complex and nonlinear [14, 15]. Such complex relations can best be investigated by taking advantage of flexible ML models. Consequently, advanced ML methods, including RFs, outperform simpler models in predicting TSH, as recently shown by Santhanam et al. [16]. In their study, the best scoring models were RF, gradient boosting and stacking regression with coefficient of determination of $R^2 = 0.13$ and a mean absolute error of 0.78. The models were based on a small set of preselected thyroid-related risk factors, including free thyroxine (FT4), free triiodothyronine (FT3), autoantibodies to thyroid peroxidase (anti-TPO), as well as Body Mass Index (BMI), age and ethnicity.

In large parts, existing related literature focuses only on the ML-based classification of thyroid disease [17, 18]. However, clinical reference ranges sometimes fail to distinguish actual disease states from ordinary fluctuations in case of complex, multifactorial diseases like thyroid dysfunction. Especially since serum TSH levels within the reference range are also known to vary by age, sex, the applied assay, and the population's background iodine status [19], labels derived from TSH alone may be imprecise. Therefore, we considered the problem as a regression problem. Nevertheless, also in the classification case, decision-tree-based algorithms were found to score superiorly [20, 21].

To date, it is still unclear how to automatize the prediction from high-dimensional clinical data, and how to present results of a complex ML model, such that it can be interpreted easily by medical professionals as well as non-experts. The state-of-the-art to interpret RFs is to use global feature importance (FI). FI measures the global influence of every individual feature on the model and may be used to create a ranking. Model internal measures may be used as FI, such as the increase in homogeneity in the trees' leaves. Apart from that, external measures are available that evaluate the FI on out-of-bag data. One such example is permutation-based FI, which was introduced initially for RFs and later generalized [22]. However, all these FI measures neglect feature interactions. Thus, the resulting ranking may suffer from disruptive effects in the presence of heterogeneity and multicollinearity, as present in SHIP data: Continuous features or features with many categories offer more flexibility and may gain higher importance than binary features. Moreover, permutation of one feature alone may result in unrealistic data instances, so associated features may bias the importance score. Lastly, indirect effects and confounders cannot be noticed from the ranking alone. Other available interpretation methods focus often only on bivariate feature associations. To take more complex feature associations into account and offer an interpretation that goes beyond a ranking, we present a workflow that complements the RF model by including a Bayesian network (BN) as surrogate model. As Bayesian network structure learning from data is highly computationally expensive, we reduce the feature set by extracting potentially relevant features from the RF model for this step. A key concept in the context of BNs is the Markov blanket (MB). The MB of a node in a BN is the set of directly dependent variables in the network, i.e., those features that are most important to predict a particular variable and have a direct influence on it. MBs are often used for selecting optimal feature subsets and they also offer an interpretation framework for the interpretation of larger BNs, that we make use of [23–26].The presented ML-based workflow allows to identify and interpret broad patterns from high-dimensional data. We apply this

workflow to predict the individual serum TSH concentration from clinical data and to identify broad and interpretable clinical patterns of thyroid functionality using data from the Study of Health in Pomerania (SHIP) [2]. The dataset includes nutritional patterns, complete blood counts, sociodemographic data, health status, mood, medication, detailed thyroid examinations, and genetic information in the form of single nucleotide polymorphisms (SNPs) of 4308 adult individuals. While many of the discovered factors have been analyzed in univariate studies before, to our best knowledge, this is the first thyroid study applying ML-based algorithms to identify multivariate patterns of such broadness.

## Results and discussion

### Workflow

We propose a workflow that complements a RF model with a BN analysis for the post-hoc interpretation of inferred global predictive patterns. The network analysis allows an interpretation that goes beyond a ranking by global feature importance.

The workflow consists of three main steps (Fig 1) and starts with careful training of a RF model. RF is an ensemble learning algorithm that aggregates multiple single decision trees. In the first step, we perform a hyperparameter optimization of the main RF parameters using a nested cross-validation approach. Afterwards, relevant features are identified from the RF model using two different FI measures. Due to the high number of features included, we use a statistical mixture model approach to distinguish relevant from irrelevant features. The latter are modeled by a component around a FI of zero. This step is followed by BN structure learning among all relevant predictors. BNs are probabilistic graphical models. The BN is trained using a score-based approach. This last step yields an interpretable feature association model. For further details on the workflow, we refer the reader to the methods section.

### Simulation study

We first evaluate the effectiveness of our workflow and the influence of parameters like sample size and feature number by a simulation study. We create synthetic models of different sizes. We observed that the correct feature subset was identified, with slightly better results for smaller networks and larger sample sizes (see Table 1). In the large network, in average 6 out of 20 nodes were not identified as related to the target (false negative; fn). This effect can be

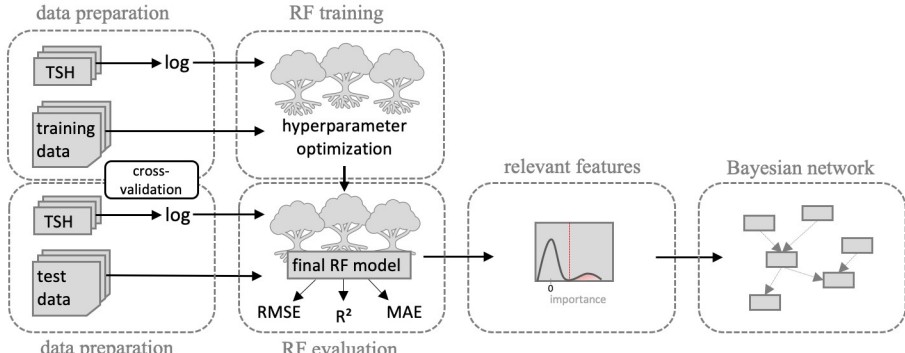

**Fig 1. Workflow.** Schematic representation of the workflow. After data preparation, a RF model is trained using nested cross-validation. Relevant predictors are identified based on two feature importance measures and a mixture model approach. Lastly, feature interactions among the relevant predictors are examined in a Bayesian network analysis.

Table 1. **The table reports the results of the simulation study.** A) Feature Subset Extraction: Average of (false positive (fp) / false negative (fn)) features (averaged over 10 iterations). B) Relearned Network Structure: Average of false positive (fp) / false negative (fn) arcs of the relearned network structure (averaged over 10 iterations).

| Network<br>Sample Size | small network (5 nodes) | medium network (10 nodes) | large network (20 nodes) |
|---|---|---|---|
| **A) Feature Subset Extraction** | | | |
| **Small (500)** | 0.4 fp / 2 fn | 0.2 fp / 1 fn | 1.6 fp / 6.2 fn |
| **Medium (1000)** | 0.4 fp / 1.6 fn | 0 fp / 1.2 fn | 1.4 fp / 5.8 fn |
| **Large (2000)** | 0.4 fp / 1.6 fn | 0.2 fp / 0.8 fn | 0.6 fp / 5.8 fn |
| **B) Bayesian Network Structure Learning** | | | |
| **Small (500)** | 0 fp / 1.2 fn | 2.4 fp / 2.4 fn | 2.6 fp / 2.8 fn |
| **Medium (1000)** | 0 fp / 1.2 fn | 1 fp / 1.2 fn | 1.8 fp / 2.2 fn |
| **Large (2000)** | 0 fp / 0.8 fn | 1 fp / 1.1 fn | 0.8 fp / 2 fn |

explained by the size of the network and their very indirect connection in the sampled models. Thus, their association to the target is very weak.

In some cases, noise variables were identified as predictors by the RF model (false positive; fp). This observation underlines the need for a more detailed analysis than single variable importance rankings. A closer look on these cases revealed that those features had no neighbors in the relearned BN. The post-hoc BN analysis thus allows to correctly identify those features as unimportant.

The relearned BN structures could in general also be correctly recovered (Table 1). We observed in all scenarios less than 3 false positive and false negative arcs. The network quality was higher for smaller networks and larger sample sizes. In general, the number of false negative arcs was higher than the number of false positive arcs. This may be due to the randomly sampled parameters with weak differences for some arcs.

## Predicting TSH concentrations

**Data description.**   The preprocessed SHIP dataset includes 602 features and 3,989 probands (49% female, 51% male, for details on data preprocessing, see Methods). The mean age of study participants is 49 years (min = 20, max = 81). In addition to serum TSH, thyroid function evaluation in SHIP includes FT3, FT4, and anti-TPO measurements. Besides, urinary iodine and enlargement of the thyroid ('goiter') were assessed. Results of a sonography examination of the thyroid were included in terms of thyroid volume, echogenicity, and the presence of at least one thyroid nodule. Descriptive statistics about these features are reported in Table 2. Additional features from SHIP include nutritional patterns, complete blood counts, sociodemographic information, health status, and medication. Moreover, we added 67 prefiltered SNPs to examine possible genetic predispositions [27].

**Random forest predictions.**   Our RF model achieves an RMSE of 0.663 (± 0.003), coefficient of determination ($R^2$) of 0.15 (± 0.002), and a mean absolute error (MAE) of 0.55 (± 0.003) on unseen data. For evaluation purposes, we compared the achieved prediction scores to a baseline model trained on the same dataset with TSH values randomly shuffled. The shuffling breaks all relations of TSH to the remaining data; thus, the baseline predictions can be interpreted as scores achieved by random guessing (Table 3).

**Extraction of relevant features.**   We identified 62 out of 602 features as relevant in the RF model. For the identification, we used two different FI measures and a statistical mixture model approach (see Methods). All relevant features are reported in S1 Table. The highest importance scores were found for age, FT3, FT4, anti-TPO antibodies, goiter, nodules, and

**Table 2. Descriptive statistics of thyroid examination results from SHIP.** Mean, standard deviation, median and skewness are presented for continuous features. For categorical features, the exact distribution is shown. The analysis is based on n = 3,989 probands.

| SHIP Variable | Description | Mean | StDev | Median | Skewness |
|---|---|---|---|---|---|
| tsh | Thyroid stimulating hormone (TSH) [mU/l] | 0.89 | 2.28 | 0.66 | 25.5 |
| log_tsh | log-transformed TSH | -0.45 | 0.73 | -0.41 | -0.65 |
| ft3 | free triiodothyronine [pmol/l] | 5.25 | 0.88 | 5.2 | 1.24 |
| ft4 | free thyroxine [pmol/l] | 12.84 | 3.82 | 12.5 | 1.24 |
| sd_volg | total sonography volume of the thyroid | 21.54 | 12.57 | 18.8 | 3.35 |
| jodid_u | Iodide (urine) [µg/dl] | 14.42 | 11.64 | 12.5 | 5.1 |
| tpo_ak | anti-TPO antibodies [IU/l] | 90.28 | 294.28 | 45.1 | 25.47 |
| SHIP Variable | Description | No | | Yes | |
| node_s0 | presence of thyroid nodule(s) | 3299 (77.2%) | | 975 (22.8%) | |
| echogenthyr_s0 | hypoechoic thyroid pattern | 3958 (92.7%) | | 313 (7.3%) | |
| goiter_s0 | enlargement of the thyroid gland | 2660 (62.2%) | | 1611 (37.8%) | |

thyroid hypoechogenicity in sonography (S1 and S2 Figs). The 62 extracted relevant predictors are distributed across the categories basic patient information (8), information about the general health status (5), thyroid examinations (8), metabolism (9), SNPs (3), socioeconomic status (8), diet (5), immune system (6), hematological and hemostasis parameters (5), hormones (3), and electrolyte levels (2).

**Bayesian network analysis.** In order to investigate the association patterns of TSH, we complement the RF model with a better interpretable BN. Whereas RFs are optimized with respect to high predictive power alone, BNs are probabilistic models of feature interactions. They allow examining how features are associated and how these associations affect the outcome. Their graphical representation allows for intuitive interpretation, also for non-experts. Based on the methodology described in an earlier study [28], we train a BN, including only those features that were identified as relevant in the RF model. To reduce the network's complexity, we aggregated highly collinear features to represent them as one single node in the network. The network structure among the resulting 54 nodes, which are reported in S1 Fig, was then learned by a score-based structure learning approach. The final Bayesian network structure (Fig 2) has 128 edges. The nodes *sex* and *age*, number of medications taken during the last seven days (*medic7d_s0*), and hip circumference (*som_huef*) are hub nodes in the network. Additionally, we identified four different clusters within the network that refer to the categories socioeconomic status, metabolism, hematological and hemostasis factors, and thyroid examinations (S1 Table). The Markov blanket of a node in a BN is the set of directly dependent variables in the network, i.e., those features that are most important to predict a particular

**Table 3. Evaluation of the final RF model for the prediction of TSH.** As a baseline comparison, we trained a similar model on the same dataset where TSH values have been randomly shuffled. The scores given in the column (random) baseline prediction thus represent scores achieved by random guessing. Average results are presented together with standard deviations given in brackets.

| Evaluation criteria | prediction of TSH (± SD) | (random) baseline prediction (± SD) |
|---|---|---|
| RMSE Training | 0.63 (± 0.041) | 0.70 (± 0.004) |
| RMSE Test | 0.66 (± 0.003) | 0.72 (± 0.301) |
| $R^2$ Training | 0.23 (± 0.003) | 0.0001 (± 0.002) |
| $R^2$ Test | 0.15 (± 0.002) | 0.0004 (± 0.011) |
| MAE Training | 0.52 (± 0.002) | 0.52 (± 0.001) |
| MAE Test | 0.55 (± 0.003) | 0.62 (± 0.111) |

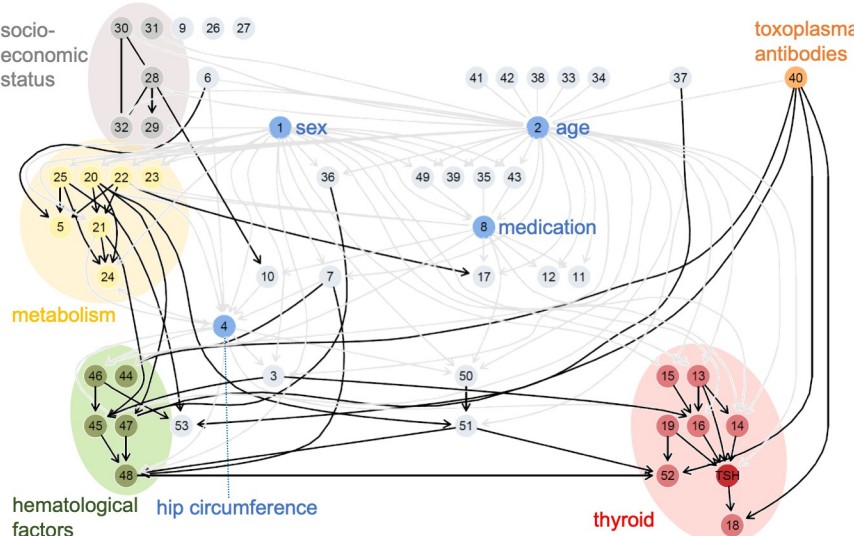

**Fig 2. Inferred Bayesian network structure among the extracted relevant predictors and the TSH level.** The four hub nodes, sex, age, medication (taken during the last seven days), and hip circumference are colored in blue. Arcs originating from the hub nodes are plotted in light gray to make the network more readable. The TSH level is colored in dark red, thyroid-related examinations in red. Yellow nodes refer to metabolic factors, green nodes to hematological and hemostasis factors, and grey nodes to socioeconomic parameters. Antibody titer against toxoplasmosis is presented in orange. Further information on the features can be found in S1 Table. The completed partially directed acyclic graph is shown.

variable and have a direct influence on it. The Markov blanket of the TSH level contains the predictors that have also been identified as the most important predictors based on global feature importance in the RF model (FT3, FT4, anti-TPO, low or anechoic thyroid in sonography, thyroid volume, toxoplasma antibodies, age). The average Markov blanket size of the whole network is 7.3, showing a relatively strong association among all predictors.

The extracted relevant features and their associations reveal broad clinical patterns of thyroid functionality. As expected, the top features include a person's age and thyroid-specific examinations (FT3 (node ID 18), FT4 (node ID 19), anti-TPO (node ID 14), and sonography results (node IDs 13 and 16), see S1 Table). The strong association of these features with the individual TSH level is also reflected by their closeness in the BN (nodes colored in red, TSH is dark red, Fig 2).

In addition to age, the set of relevant features includes a set of further general patient information and clinical measures (e.g., sex, body height, hip circumference, heart rate, blood pressure) as well as measures describing the health status of a person (number of doctoral visits, subjective physical and mental health, medication; node IDs 1–12). As expected, sex and age influence nearly every other feature; they are hub nodes in the Bayesian network, as are the amount of medication and hip circumference (blue nodes, Fig 2).

Moreover, eight of the 62 relevant features describe a person's socioeconomic status and are related to occupation, education, or family status (node IDs 28–32). The BN shows that they are closely tied to a person's age and health status so that their influence on the TSH level is presumably only indirect (nodes colored in gray, Fig 2).

The same holds for most included dietary factors (amount of grey bread, cake, fresh fruits; IDs 33–35) that we found mainly related to health status and age. On the contrary, we found the daily amount of alcohol (node ID 36) influencing the mean corpuscular volume and the liver status, both well-established markers of alcohol use [29]. Alcohol consumption is thereby

indirectly linked to TSH. Also, coffee consumption (node ID 37) was indirectly linked via the serum potassium level. Its association with the thyroid was studied extensively in targeted studies [30, 31].

Moreover, factors related to the status of liver and kidney, hematological and hemostasis status, immunity, hormones, lipid and glucose metabolism, as well as electrolyte levels have been identified as relevant (node IDs 20–25). Three of the 67 SNPs are included as well; two of them are highly related and occur in the *phosphodiesterase 8B* gene. They have been combined to one node in the network (node 26). The third one is a variant near *FOXE1* (node 27), which is also known as thyroid transcription factor 2. All three have been associated with altered TSH levels in earlier studies [27]. The active inclusion of SNPs into the RF model affirms the genetic component of thyroid (dys-) function.

The association of TSH with liver and kidney markers can be explained physiologically. It is well known that thyroid hormones affect renal physiology, hepatic function, and bilirubin metabolism. We here identified an altered glomerular filtration rate, altered serum creatinine levels, and levels of serum uric acid as associated with TSH. A correlation in the case of healthy as well as diseased thyroid states has been observed before [32, 33]. With thyroid hormones regulating the basal metabolic rate of hepatocytes, it is no surprise that changes in the thyroid homeostatic system also go along with hepatic disorders. In our model, the presence of hepatic steatosis, serum aspartate-aminotransferase levels, ferritin levels, but also serum glucose, lipase, and triglycerides appear to be relevant predictors. While an association of liver and thyroid disease has been examined for a long time, it is still under debate if this correlation is independent of the metabolic syndrome or can be fully explained by alterations in glucose and lipid metabolism [34–37].

In the BN, the correlation between metabolic measures (nodes colored in yellow, Fig 2) and thyroid is predominantly explained by health status and hematological and hemostasis parameters (nodes colored in green, Fig 2). Hematological parameters (like mean corpuscular volume, hematocrit, number of leukocytes; node IDs 45, 46, 48) and hemostasis parameters (like partial thromboplastin time and fibrinogen; node IDs 44 and 46) are widespread general measures for health and disease. However, it is also well-known that overt hypothyroidism is associated with a bleeding tendency, while hyperthyroidism leads to increased coagulation and decreased fibrinolysis. Recent studies suggest that coagulation factors have a mediating role between thyroid and cardiovascular abnormalities [38, 39]. Our Bayesian network model is in line with this hypothesis, as it links metabolic factors to thyroid hormones mainly via hematological and hemostasis factors. However, more targeted studies are needed to evaluate this association further.

Additionally, several included features are related to vaccinations or infections, including immunity against rubella, measles, toxoplasmosis, or *Helicobacter pylori* (node IDs 38–43). Most of them appear to be primarily correlated to age with only minor independent effects on TSH (e.g., insufficient immunization against rubella or measles is more frequent in older people). However, the level of antibodies against toxoplasmosis (node colored in orange, Fig 2) is strongly related to thyroid hormone levels, predominantly to free T3. The association between *Toxoplasma gondii* infections and thyroid dysfunction was observed in earlier studies as well, underlining the role of toxoplasmosis antibodies as an independent predictor of thyroid hormone levels [40, 41].

Thyroid function also plays an essential role in the balancing of sex hormones. We identified serum prolactin, sex hormone-binding globulin, and insulin-like growth factor 1 (IGF-1) levels as predictors of TSH. Already earlier, subclinical hypothyroidism was shown to increase the prevalence of overt hyperprolactinemia, particularly in women. It was also shown to indirectly influence IGF-1 and sex hormone-binding globulin (SHBG) levels [42–44].

**Table 4. RF prediction results for different feature subgroups.** Columns refer to models built based on different feature subgroups. The first two rows show the respective RF hyperparameters. The remaining six rows contain the prediction metrics achieved by the models. Average results are stated with standard deviations given in brackets.

| Model | (random) baseline prediction (± SD) | All Features (± SD) | metabolism [yellow nodes] (± SD) | socioeconomic status [grey nodes] (± SD) | hematological factors [green nodes] (± SD) |
|---|---|---|---|---|---|
| RMSE Training | 0.703 (± 0.003) | 0.632 (± 0.003) | 0.697 (± 0.003) | 0.702 (± 0.003) | 0.702 (± 0.003) |
| RMSE Test | 0.719 (± 0.029) | 0.662 (± 0.032) | 0.703 (± 0.003) | 0.704 (± 0.032) | 0.705 (± 0.032) |
| MAE Training | 0.599 (± 0.104) | 0.515 (± 0.11) | 0.592 (± 0.105) | 0.598 (± 0.104) | 0.598 (± 0.104) |
| MAE Test | 0.618 (± 0.106) | 0.551 (± 0.11) | 0.599 (± 0.111) | 0.601 (± 0.111) | 0.601 (± 0.111) |
| $R^2$ Training | 0.045 (± 0.008) | 0.229 (± 0.0035) | 0.061 (± 0.002) | 0.046 (± 0.002) | 0.046 (± 0.002) |
| $R^2$ Test | -0.0004 (± 0.002) | 0.149 (± 0.023) | 0.042 (± 0.021) | 0.037 (± 0.015) | 0.037 (± 0.015) |

**Random forest prediction from feature subgroups.** To complement the analysis, we examined how well a RF can predict individual TSH concentrations from one of the identified feature subgroups 'metabolism', 'socioeconomic status', and 'hematological factors' alone. Table 4 reports the performance scores of predicting TSH measures when a new RF model was trained using only sex, age, and features from one category. The best performance was achieved in case of the feature subgroup 'metabolism' (Test RMSE of 0.703). However, all metrics and (especially $R^2$) decreased considerably due to the reduced feature sets.

## Conclusion

In summary, we demonstrate that the combination of RF and BN analysis is useful to reveal and interpret broad association patterns. In our work, we showed that a complementary network analysis can overcome classical drawbacks of RF interpretation based on feature importance only and is helpful for high-quality interpretation. We tested the workflow using simulated data and applied it to real data from SHIP. Based on the promising findings presented in this paper, future research directions may include to follow up other predictive models than RF by BN analyses.

The identified predictive patterns of TSH concentrations are in line with recent findings and give new insights into thyroid functionality. The most important predictors included a person's age and thyroid-specific parameters: FT3, FT4, anti-TPO, and sonography results (S1 Fig). It must be noted that relevant predictors were successfully revealed automatically from an extensive set of features. Yet, the presented model yields better prediction results than models built from a small, manually chosen feature set. Sex, age, hip circumference, and medication intake during the last seven days were further identified as hub nodes in the BN of relevant predictors. Based on the network, clusters of related features could be identified and further tested for their predictive capacity. However, a large fraction of variance in the data remains unexplained. Possibly, parts of the leftover variance are due to temporal fluctuations. The inclusion of individual temporal profiles, or at least measurements at more than one time point, would mark an interesting extension and could further increase the prediction accuracy. Due to the large number of features, the final model lacks validation on external data. Such validation is challenging, as an appropriate dataset would need to include all of the used features with similar measurement protocols. However, the discovered patterns are well supported by state-of-the-art literature.

In contrast to often-used classification models, the presented regression model is independent of clinical TSH reference limits or the diagnosis of specific dysfunctions. Thus, it may also be used to detect disease initiation or minor abnormalities. Our study underlines the need

for careful interpretation of complex models. It shows that a ranking by global feature importance is not enough to interpret intricate predictive patterns and can be misleading. The identified association patterns may be useful for future thyroid research and improved dosing of therapeutics.

## Materials and methods

### Study population

The Study of Health in Pomerania (SHIP) is a population-based study carried out in West Pomerania, the north-east area of Germany [2, 45]. A sample from the population aged 20 to 79 years was drawn from population registries. First, the three cities of the region (with 17,076 to 65,977 inhabitants) and the 12 towns (with 1,516 to 3,044 inhabitants) were selected, and then 17 out of 97 smaller towns (with less than 1,500 inhabitants) were drawn at random. Second, from each of the selected communities, subjects were drawn at random, proportional to the population size of each community and stratified by age and gender. Only individuals with German citizenship and main residency in the study area were included. Finally, 7,008 subjects were sampled, with 292 persons of each gender in each of the twelve five-year age strata. In order to minimize dropouts by migration or death, subjects were selected in two waves. The net sample (without migrated or deceased persons) comprised 6,267 eligible subjects. Selected persons received a maximum of three written invitations. In case of non-response, letters were followed by a phone call or by home visits if contact by phone was not possible. The SHIP population finally comprised 4,308 participants (corresponding to a final response of 68.8%).

### Ethics statement

The SHIP study was approved by the ethics committee of University Medicine Greifswald, approval number BB 174/15. Written informed consent was obtained from all study participants.

### Data preprocessing

In addition to phenotypical features from SHIP (including nutritional patterns, complete blood counts, sociodemographic data, health status, mood, medication, and detailed thyroid examinations), additional data about the presence or absence of 67 SNPs were included in our analysis, that have previously been shown to be associated with thyroid dysfunction in a genome-wide association study (GSWA) [27]. From the original dataset, features with more than 20% of missing values were removed, and the remaining data were imputed using a non-parametric, RF-based imputation procedure [46]. We further removed participants under anti-thyroid medication (n = 280) and those for which the information about anti-thyroid medication was missing (n = 37). We also removed participants with extremely high TSH measurements that exceeded 60 mU/l (n = 2). In total, 3989 participants and 602 features (67 SNPs) were used for further analysis. As the distribution of TSH concentrations is heavily right-skewed, we log-transformed the target variable for further analysis to make its distribution more symmetric.

### Genotyping

Non fasting blood samples were drawn from the cubital vein in the supine position. The samples were taken between 07:00 AM and 04:00 PM, and serum aliquots were prepared for immediate analysis and storage at -80˚C in the Integrated Research Biobank (Liconic, Liechtenstein). The SHIP samples were genotyped using the Affymetrix Genome-Wide Human

SNP Array 6.0. Hybridization of genomic DNA was done following the manufacturer's standard recommendations. Genetic data were stored using the database Caché (InterSystems). Genotypes were determined using the Birdseed2 clustering algorithm. For quality control purposes, several control samples were added. On the chip level, only subjects with a genotyping rate on QC probe sets (QC call rate) of at least 86% were included. Finally, all arrays had a sample call-rate of above 92%. The overall genotyping efficiency of the GWA was 98.55%. Imputation of genotypes in SHIP was performed using the software IMPUTE v2.2.2 based on the 1000 Genomes release Mar 2012 ALL populations reference panel. SNPs with a Hardy-Weinberg-Equilibrium p-value <0.0001 or a call rate <0.8 were removed before imputation.

## Evaluation criteria

To evaluate the predictive capacity of a model, we use the root mean square error (RMSE), coefficient of determination ($R^2$), and mean absolute error (MAE). The MAE is the mean of the absolute differences between data and predictions. It is non-negative, with a value of zero indicating a perfect prediction. Its value is dependent on the scale of the outcome variable. Similarly, the RSME can be calculated as the quadratic mean of these differences. Lastly, the coefficient of determination measures the proportion of the variance that is explained by the model; it ranges from 0 to 1.

## Random forests

RFs are ensemble models that combine a multitude of decision trees [1]. They output the mean prediction of the individual trees, which are guaranteed to be decorrelated due to the use of bootstrap samples and random feature subsets of the training data. RFs are considered black-box models, as it is very difficult to retrace how the model came to a specific prediction. In order to reduce the bias in model selection, we applied nested 10-fold cross-validation for hyperparameter optimization. The results from hyperparameter optimization and the final parameter settings are reported in S3 Fig and S2 Table. For training of the RF model, we used the R-package randomForest [47]. We optimize the three main hyperparameters, which control the structure and depth of the forest, based on internal 10-fold cross-validation in a grid-search: the minimum size of terminal nodes (*nodesize*, tested values: 15, 40, 65), the maximal number of terminal nodes (*maxnodes*, tested values: 15, 40, 65), and the number of variables randomly sampled as candidates at each split (*mtry*, tested values 3–50). External cross-validation was then applied to evaluate the performance of the final model on unseen data. The RSME was chosen as an objective function; additionally, MAE and $R^2$ of a model are used as evaluation criteria.

## Measures of feature importance

Feature importance in the RF model was assessed using two different measures: node purity and incremental mean square error (IncMSE). Node purity measures the increase in homogeneity (here in terms of variance) of the labels at the respective node. The final node purity value of a feature is the sum over all splits in which the feature is chosen, averaged over all trees. Conversely, the IncMSE is a permutation-based FI measure, and it is calculated as the difference in the overall out-of-bag error before and after permutation of the feature (S1 Fig).

## Extraction of relevant predictorsss

Due to the high number of included features, likely, most of them are not relevant for the prediction of TSH in the RF model. Thus, many features have importance scores close to zero S1

Fig. That is why we used an approach based on linear mixture models to distinguish between relevant and irrelevant features. We assume that irrelevant features are closely distributed around zero, with a small amount of variation due to the inherent randomness of the FI measures. In the case of the IncMSE, we used a normal distribution around zero to model all irrelevant features (S2A Fig). As the node purity takes only positive values, we assume it to be gamma-distributed instead (S2B Fig). We consider those features as relevant for which the mean importance is larger than the respective 0.999-quantile, which means they most likely stem from the set of features with importance greater than zero. Fitting resulted in a mean of 0.1 and a standard deviation of 0.85 for the component around zero in case of IncMSE, and shape parameter 0.62 and scale parameter 4.65 for the component around zero in case of node purity.

## Bayesian network analysis

We additionally analyze the feature interrelations of relevant features using a BN approach. BN are a prominent tool for probabilistic reasoning in artificial intelligence, and they model the joint distribution of a feature set in terms of a directed acyclic graph. In the graph, the nodes refer to the features (resp. random variables) $X_1,\ldots,X_n$ and arcs model conditional (in) dependencies among them. The joint distribution factorizes efficiently according to the graph structure, allowing for efficient computation and approximate inference. It can be evaluated based on local probabilities depending only on a node's parents in the graph:

$$P(X_1, \ldots, X_n) = \prod_{i=1}^{n} P(X_i | \boldsymbol{parents}(X_i)) \tag{1}$$

The visualization in terms of a network is intuitive even for non-experts and supports human interpretation. Thus, a BN structure can help to understand complex associations, identify confounding factors, and make multicollinearity visible. We use a score-based hill-climbing approach to learn a conditional Gaussian Bayesian network that models heterogeneous data [48, 49]. For the learning of the BN, we used the Bayesian information criterion (BIC) as objective function, which is a penalized likelihood criterion. We also apply a model averaging approach to reduce false positive arcs and includes the determination of arc strength [50]. Arc directions can only partly be identified by structure learning algorithms, as several network structures are equivalent in terms of the probability distribution they model. Therefore, BN structures are often visualized as completed partially directed acyclic graphs (CPDAG) that have directed arcs only for those arcs which direction is determined. Fig 2 shows the CPDAG of the inferred BN structure. For the determination of BN parameters, a completely directed model has to be chosen. That, and the network's complexity, is why we focus only on its structure.

Before learning the network, we aggregate highly collinear features based on feature similarity for heterogeneous variables to reduce the network's complexity [49, 51]. A list of features, including aggregated features, is given in S1 Table.

## Simulation study

We begin by randomly sampling three BN structures (small: 5 nodes, medium. 10 nodes, large: 20 nodes, see Table 5) consisting of a target variable and several features. We randomly sample (Gaussian) parameters depending on each node's parents. We then add noise variables as isolated nodes, that are not connected to the target or the network. Next, we generate data from the BN. We then apply our workflow (Fig 1) to the simulated data and analyze whether

**Table 5. Properties of three Bayesian network structures used for the simulation study.**

| network | nodes | arcs | parameters | additional noise variables |
|---------|-------|------|------------|----------------------------|
| small | 5 | 5 | 15 | 5 |
| medium | 10 | 23 | 43 | 10 |
| large | 20 | 26 | 66 | 20 |

(i) the correct feature set could be identified and (ii) the correct network structure can be relearned. We compare the extracted feature set and determine the number of false positive (fp) and false negative (fn) features. We also compare the relearned network structure to the according true subgraph and determine the number of false positive and false negative arcs.

## Computations

We report the different R packages that were used in our analysis with their version in S3 Table.

## Supporting information

**S1 Fig. Variable importance measures of the top 20 features from the random forest model.** Based on A) incremental mean square error (IncMSE) and B) Node purity, the highest importance scores were found for age, FT3, FT4, anti-TPO antibodies, goiter, thyroid nodules, and thyroid hypoechogenicity in sonography. The important features were extracted from random forest based on the selected parameter nodesize of 15, and maxnodes of 15.
(TIF)

**S2 Fig. Estimated density of feature importance measures.** A statistical mixture model was used, the component around zero (red dashed lines) was modeled as A) a normal distribution for IncMSE (mean 0.1, standard deviation) B) a Gamma distribution for node purity (shape parameter p = 0.62 and scale parameter b = 4.65). Features were identified as relevant if they had feature importance larger than the respective 0.999-quantile.
(TIF)

**S3 Fig. Hyperparameter optimization.** Prediction results from the grid-based hyperparameter optimization of the random forest model using 10-fold nested cross-validation. Rows show results for varying values of the parameter nodesize (tested values: 15, 40, 65), and columns show results for varying values of the maximal number of terminal nodes (maxnodes, tested values: 15, 40, 65). The parameter mtry was optimized separately and set to 5.
(TIF)

**S1 Table. List of SHIP variables that have been identified as relevant in the RF model.** The IDs refer to nodes in Fig 2, which are categorized into basic information, general health status, Thyroid examination, Metabolism, SNPs, Socioeconomic Status, Diet, immune system, Hematological and Hemostasis parameters, Hormones and Electrolytes. The relevant features from RF model are described in third column.
(XLSX)

**S2 Table. Hyperparameters of a random forest model and their description.** This table reports the different RF hyperparameters that were optimized, with describing the features in second column. The third column contains the parameter ranges for parameters for optimization. The last column contains the final parameter selected for final RF model presented

above.
(XLSX)

**S3 Table. R Software packages and versions.**
(XLSX)

**S1 File. R software code of simulation analysis.**
(R)

## Author Contributions

**Conceptualization:** Ann-Kristin Becker, Till Ittermann, Lars Kaderali, Neetika Nath.

**Data curation:** Ann-Kristin Becker, Till Ittermann, Markus Dörr, Stephan B. Felix, Matthias Nauck, Alexander Teumer, Uwe Völker, Henry Völzke, Neetika Nath.

**Formal analysis:** Ann-Kristin Becker, Lars Kaderali, Neetika Nath.

**Funding acquisition:** Lars Kaderali.

**Investigation:** Ann-Kristin Becker, Till Ittermann, Neetika Nath.

**Methodology:** Ann-Kristin Becker, Lars Kaderali, Neetika Nath.

**Project administration:** Lars Kaderali.

**Resources:** Till Ittermann, Markus Dörr, Stephan B. Felix, Matthias Nauck, Alexander Teumer, Uwe Völker, Henry Völzke, Lars Kaderali.

**Software:** Ann-Kristin Becker, Neetika Nath.

**Supervision:** Lars Kaderali.

**Validation:** Ann-Kristin Becker, Till Ittermann, Markus Dörr, Alexander Teumer, Neetika Nath.

**Writing – original draft:** Ann-Kristin Becker, Neetika Nath.

**Writing – review & editing:** Till Ittermann, Markus Dörr, Stephan B. Felix, Matthias Nauck, Alexander Teumer, Uwe Völker, Henry Völzke, Lars Kaderali.

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
