## [Decision Letter · Decision Letter 0]

13 Apr 2022

PONE-D-22-03831Analysis of epidemiological association patterns of serum Thyrotropin by combining random forests and Bayesian networksPLOS ONE

Dear Dr. Kaderali,

Thank you for submitting your manuscript to PLOS ONE. After careful consideration, we feel that it has merit but does not fully meet PLOS ONE’s publication criteria as it currently stands. Therefore, we invite you to submit a revised version of the manuscript that addresses the points raised by the reviewers during the review process.

We look forward to receiving your revised manuscript.

Kind regards,

Holger Fröhlich

Academic Editor

PLOS ONE

Journal Requirements:

2. We noted in your submission details that a portion of your manuscript may have been presented or published elsewhere.

(Full submission to PLOS Computational Biology, This is a transfer submission.) 

Please clarify whether this publication was peer-reviewed and formally published. If this work was previously peer-reviewed and published, in the cover letter please provide the reason that this work does not constitute dual publication and should be included in the current manuscript.

Reviewers' comments:

Reviewer's Responses to Questions

**Comments to the Author**

1. Is the manuscript technically sound, and do the data support the conclusions?

Reviewer #1: Yes

Reviewer #2: Yes

2. Has the statistical analysis been performed appropriately and rigorously? 

Reviewer #1: Yes

Reviewer #2: Yes

3. Have the authors made all data underlying the findings in their manuscript fully available?

Reviewer #1: Yes

Reviewer #2: Yes

4. Is the manuscript presented in an intelligible fashion and written in standard English?

Reviewer #1: Yes

Reviewer #2: Yes

5. Review Comments to the Author

Reviewer #1: The manuscript " Analysis of epidemiological association patterns of serum Thyrotropin by combining random forests and Bayesian networks " presents a valuable and interesting approach to how two techniques can be combined to create predictive models. The work has been carried out with care and is presented in sufficient detail. The manuscript has been reviewed before and thereafter transferred to PlosONE. Reviser’s comments, re-comments by the authors and corrections in the text are contained in the submission. The editor comment on rejection was: ”While the reviewers do not find any major technical flaws that would preclude publication of the work, they also mark on the lack of depth of the manuscript and potential practical relevance”.

I agree with the editors statement, that there are no major technical flaws that would preclude publication. I disagree with the statement of “lack of depth of the manuscript and potential practical relevance”.

Main critics was given by previous reviewer #1 in two points:

a) He/she is missing a comparison with other existing methods … making this research of interests for the bioinformatics and ML community.

The reviewer's request is understandable but hardly realizable. There a plenty of ways how build reliable prediction models, especially if techniques are combines. Further, the generalization of any comparison lacks the scenarios simulated for it. The authors already present a simulation, showing that their approach picks out necessary features for prediction. They intensively applied cross validation, a well-accepted technique for robust model building.

b) The reviewer implicitly focus on ML models.

I understand the author’s aim of their approach is in improving prediction by combining methods, not stating that their approach is universal best. It goes without saying, that the concept can also be used if, for example, the Bayesian Network is replaced by a ML-prediction models.

c) Another point of critics was given by previous reviewer #2 regarding the practical value of predicting TSH levels from other blood-based variables, instead of directly measuring from blood.

A possible application of the presented results could be follows: A measured TSH level that exceeds the predicted value may raise medical concerns about finding the cause. Or: A predicted high TSH value should be verified by direct measurement. The application of the proposed results is not the scope of the manuscript.

All in all, I recommend the manuscript for publication without the need for further revision.

Reviewer #2: Notes:

1. The algorithm as outlined is reasonable but the approach is not entirely novel. The authors perform what's known as Markov Blanket feature selection before modeling the data with a Bayes Net. Indeed, it is somewhat unusual to perform BN fitting without prior variable selection on a large-enough dataset. For a similar example where SVMs are used to perform feature selection see, for example:

Shen et al, Markov Blanket Feature Selection for Support Vector Machines, AAAI, 2008

and for the theoretical justification of this type of a first-pass filter see this foundational paper:

Koller and Sahami, Toward Optimal Feature Selection, ICML, 1996

This comment should be interpreted as a suggestion for a useful reference or two, which would help build the context for the work, but does not invalidate the technical accuracy or practical utility of the authors' approach. Please consider this as a request for a very minor revision.

2. Please note in the Results and Discussion paper that all of the algorithm details are specified in the Methods section. While that's a reasonable place to expect those things to be, and while the level of detail in that section is adequate, the paper is sufficiently technical that I've asked myself what the RF- and BN-fitting details might be at least 20 times before finally reaching the methods section. It might even be worth it to invest a sentence per algorithm to give at least a high-level outline of the methods the first time they are mentioned in Results.

6. PLOS authors have the option to publish the peer review history of their article (what does this mean?). If published, this will include your full peer review and any attached files.

Reviewer #1: **Yes: **Albert Rosenberger

Reviewer #2: **Yes: **Boris Hayete

---

## [Author Response · Author response to Decision Letter 0]

4 Jul 2022

We thank the reviewers for their positive feedback. Please see the "Response to Reviewers" letter for a detailed response to the individual points raised.

---

## [Editor Report · Decision Letter 1]

5 Jul 2022

Analysis of epidemiological association patterns of serum Thyrotropin by combining random forests and Bayesian networks

PONE-D-22-03831R1

Dear Dr. Kaderali,

We’re pleased to inform you that your manuscript has been judged scientifically suitable for publication and will be formally accepted for publication once it meets all outstanding technical requirements.

Kind regards,

Holger Fröhlich

Academic Editor

PLOS ONE
---

## [Editor Report · Acceptance letter]

12 Jul 2022

PONE-D-22-03831R1 

Analysis of epidemiological association patterns of serum Thyrotropin by combining random forests and Bayesian networks 

Dear Dr. Kaderali:

I'm pleased to inform you that your manuscript has been deemed suitable for publication in PLOS ONE. Congratulations! Your manuscript is now with our production department. 

Kind regards, 

on behalf of

Prof. Dr. Holger Fröhlich 

Academic Editor

PLOS ONE